# Mechanistic Analysis of CCP1 in Generating ΔC2 α-Tubulin in Mammalian Cells and Photoreceptor Neurons

**DOI:** 10.3390/biom13020357

**Published:** 2023-02-12

**Authors:** Takashi Hotta, Alexandra Plemmons, Margo Gebbie, Trevor A. Ziehm, Teresa Lynne Blasius, Craig Johnson, Kristen J. Verhey, Jillian N. Pearring, Ryoma Ohi

**Affiliations:** 1Department of Cell and Developmental Biology, University of Michigan, Ann Arbor, MI 48109, USA; 2Department of Ophthalmology, University of Michigan, Ann Arbor, MI 48109, USA

**Keywords:** tubulin, detyrosination, ΔC2-tubulin, VASOHIBIN, cytosolic carboxypeptidase (CCP)

## Abstract

An important post-translational modification (PTM) of α-tubulin is the removal of amino acids from its C-terminus. Removal of the C-terminal tyrosine residue yields detyrosinated α-tubulin, and subsequent removal of the penultimate glutamate residue produces ΔC2-α-tubulin. These PTMs alter the ability of the α-tubulin C-terminal tail to interact with effector proteins and are thereby thought to change microtubule dynamics, stability, and organization. The peptidase(s) that produces ΔC2-α-tubulin in a physiological context remains unclear. Here, we take advantage of the observation that ΔC2-α-tubulin accumulates to high levels in cells lacking tubulin tyrosine ligase (TTL) to screen for cytosolic carboxypeptidases (CCPs) that generate ΔC2-α-tubulin. We identify CCP1 as the sole peptidase that produces ΔC2-α-tubulin in *TTLΔ* HeLa cells. Interestingly, we find that the levels of ΔC2-α-tubulin are only modestly reduced in photoreceptors of *ccp1^−/−^* mice, indicating that other peptidases act synergistically with CCP1 to produce ΔC2-α-tubulin in post-mitotic cells. Moreover, the production of ΔC2-α-tubulin appears to be under tight spatial control in the photoreceptor cilium: ΔC2-α-tubulin persists in the connecting cilium of *ccp1^−/−^* but is depleted in the distal portion of the photoreceptor. This work establishes the groundwork to pinpoint the function of ΔC2-α-tubulin in proliferating and post-mitotic mammalian cells.

## 1. Introduction

Microtubules are important for a vast array of cellular processes, including intracellular transport, segregating chromosomes during cell division, and formation of motile and primary cilia [1]. The functional plasticity of microtubules derives in part from their post-translational modifications (PTMs), which include phosphorylation, acetylation, methylation, glutamylation, and glycylation [2,3]. These PTMs occur through covalent modification by “writer” enzymes, but the tyrosinated (Y) C-terminal tail (CTT) of α-tubulin can also undergo enzymatic shortening by specific exo-peptidases to sequentially generate detyrosinated (Δ1 or ΔY), ΔC2 (Δ2), and ΔC3 (Δ3) α-tubulin [4,5,6]. Tubulin PTMs alter the structural and chemical properties of α,β-tubulin and can thus, in principle, impact the ability of microtubules to engage non-motile microtubule-associated proteins (MAPs) and molecular motor proteins, i.e., kinesins and dynein. Consistent with this idea, dynein–dynactin lands preferentially on Y-microtubules via a CAP-Gly protein motif in p150^Glued^ [7]. Conversely, kinesin-1 and CENP-E preferentially engage and translocate on ΔY-microtubules [8,9]. More recent work has demonstrated that microtubules assembled from recombinant Y, ΔY, ΔC2, or ΔC3 α-tubulin exhibit similar dynamic properties in the absence of MAPs, suggesting that PTM-specific microtubule dynamics are generated by effector (“reader”) proteins. This idea is supported by the knowledge that Y-microtubules are more effective at recruiting CLIP-170 and EB1 to microtubule plus ends, leading to preferential regulation of Y-microtubule dynamics [10].

Although microtubule detyrosination was discovered decades ago [11,12], the enzymes responsible for this modification were identified only recently [13,14,15]. Vasohibins (VASH1 and VASH2), in complex with small vasohibin-binding protein (SVBP), generate most of the ΔY-microtubules found in cultured cells [13,15] and post-mitotic cells such as neurons [16]. A second peptidase, microtubule-associated tyrosine carboxypeptidase (MATCAP), was recently discovered to also have detyrosinase activity [14]. Deletion of both classes of peptidases abolishes detyrosination in mice, suggesting that we now have complete knowledge of all enzymes that can detyrosinate microtubules. Detyrosination can be reversed by tubulin tyrosine ligase (TTL; [17,18,19]), but further processing of α-tubulin to the ΔC2 and ΔC3 forms is thought to be irreversible [2]. Deletion of TTL in mice thus leads to widespread increases in the levels of ΔY- and ΔC2-α-tubulin [20]. Although ΔY-microtubules are more stable than Y-microtubules, owing partly to their resistance to depolymerization by the kinesin-13s MCAK and KIF2A [21], TTL knockout mice survive to birth but die shortly after due to defects in neuronal organization [20,22]. Enzymes that generate ΔC2 and ΔC3 α-tubulin are much less well understood. Cytosolic carboxypeptidases (CCPs) have been investigated as candidate enzymes that hydrolyze the −1 and −2 glutamate residues (E) from the α-tubulin CTT (α-CTT). Of the six encoded by the human genome, overexpression of five CCPs (CCP1, CCP2, CCP3, CCP4, and CCP6) can generate ΔC2-α-tubulin in cells [6,23]. However, we do not know which CCP(s) is responsible for producing ΔC2-α-tubulin under native conditions or in the absence of TTL. Depletion of CCP1 has been shown to reduce the levels of ΔC2-α-tubulin in HEK293T cells [24], but this cell line has very low levels of ΔC2-α-tubulin (this work), so how these data relate to other cell types is unclear.

An additional factor that complicates an assessment of the biological activity and functions of the CCPs is that they have additional roles in sculpting the PTM state of both α- and β-tubulin CTTs. In addition to hydrolyzing E residues from the primary amino acid chain of α-tubulin, CCPs also remove E residues from poly-glutamate chains that are appended onto the γ-carboxyl groups of E residues in the α-tubulin CTT [6]. Defects in the poly-glutamate chain’s shortening activity of CCP1 are thought to cause degeneration of several neuronal sub-types, including Purkinje cells and photoreceptor neurons [6,25,26]. Indeed, CCP1 is the causative mutation in the mouse mutant Purkinje cell degeneration (*pcd*) [27]. Data supporting the importance of poly-glutamate chain length regulation in neuronal health include the finding that deletion of the poly-glutamylase TTLL1 suppresses Purkinje cell degeneration in *ccp1* mice [25]. As most published work of CCP1 has focused on its role in suppressing hyper-glutamylation of α-tubulin, the biological importance and subcellular localizations of ΔC2-α-tubulin have been understudied. In addition, some evidence indicates that other CCPs may act redundantly with CCP1 to generate ΔC2-α-tubulin [6]. Indeed, other CCPs are frequently co-expressed with CCP1 in most tissues [23], but the relevance of combinatorial control in generating ΔC2-α-tubulin is not known.

In this work, we identify CCP1 as the sole peptidase that generates ΔC2-α-tubulin in HeLa cells lacking TTL; HeLa cells lacking both CCP1 and TTL do not produce ΔC2-α-tubulin and contain only Y- and ΔY-microtubules. By screening a panel of cell lines, we show that ΔC2-α-tubulin is typically present at low to undetectable levels, even though CCP1 is present. Exposure of cells to taxol, a treatment that causes accumulation of ΔY-microtubules [28], does not increase the levels of ΔC2-α-tubulin. We further demonstrate that CCP1 requires ΔY-microtubules as a substrate in cells. Lastly, we provide evidence that ΔC2-α-tubulin is produced through two spatially distinct mechanisms in photoreceptor neurons. Photoreceptor neurons from *ccp1* mice lack ΔC2-α-tubulin in the distal portion of the axoneme, but ΔC2-α-tubulin is present in the connecting cilium. Collectively, our work demonstrates that ΔC2-α-tubulin is produced by CCP1 in cultured cells that are unable to revert ΔY-α-tubulin back to Y-α-tubulin but that additional peptidase(s) may function in post-mitotic cells to generate ΔC2-α-tubulin in specific cellular compartments.

## 2. Materials and Methods

### 2.1. Tissue Culture

Cultured cells were grown in the following media: CHL-1, HEK293T, U-2 OS, HeLa Kyoto, HeLa VASH1/SVBP [29], cTT20.11, HeLa *TTLΔ*, HeLa *TTLΔ CCP1Δ*—DMEM; DLD-1—RPMI-1640. Media were supplemented with 10% FBS and penicillin/streptomycin (P/S). FBS used to culture cTT20.11, HeLa *TTLΔ*, and HeLa *TTLΔ CCP1Δ* cells was tetracycline-negative to ensure suppression of Cas9 expression.

### 2.2. Animals

The *ccp1* (PCD-5J) mice were a generous gift from Albert La Spada (University of California Irvine School of Medicine Irvine, CA, USA). The *ccp1* colony was maintained by breeding *ccp^+/−^* males and females that produced *ccp1^+/+^*, *ccp1^+/−^*, and *ccp1^−/−^* pups. Mouse genotypes were determined using real-time PCR with specific probes designed for the PCD-5J strain (Transnetyx, Cordova, TN, USA). Only *ccp1^+/+^* and *ccp1^−/−^* pups were used for analysis. All mice were handled following protocols approved by the Institutional Animal Care and Use Committees at the University of Michigan (registry number A3114-01). The University of Michigan is an AAALAC accredited organization. As there are no known gender-specific differences related to vertebrate photoreceptor development and/or function, male and female mice were randomly allocated to experimental groups. All mice were housed in a 12/12 h light/dark cycle with free access to food and water.

### 2.3. Molecular Biology

Oligonucleotide DNA primers used in this study are summarized in Appendix A. cDNAs for CCPs 1, 2, 3, and 5 were obtained from Horizon discovery: CCP1 (AGTPBP1) isoform 2 (UniProt, Q9UPW5-2, 1186 aa), clone ID 30344122; CCP2 (AGBL2) isoform 1 (UniProt Q5U5Z8-1, 902 aa), clone ID 5295953; CCP3 (AGBL3) isoform 2 (UniProt Q8NEM8-2, 621 aa), clone ID 5267615; CCP5 (AGBL5) isoform 2 (UniProt Q8NDL9-2, 816 aa), clone ID, 3918769. cDNAs for CCPs 4 and 6 were synthesized by gBlocks (Integrated DNA Technologies, Coralville, IA, USA): CCP4 (AGBL1) (isoform 1, UniProt, Q96MI9, 1112 aa); CCP6 (AGBL4) isoform 1 (Uniprot Q5VU57-1, 503 aa). Each CCP cDNA was cloned into pPA-EGFP-C1 vector harboring N-terminal PA (GVAMPGAEDDVV) and mEGFP tags [30]. cDNAs were PCR-amplified using oligonucleotide primers TH738 and TH739 (CCP1), TH812 and TH813 (CCP2), TH817 and TH818 (CCP3), TH888 and TH889 (CCP4 N-terminal half, 1500 nucleotides), TH890 and TH891 (CCP4 C-terminal half, 1839 nucleotides), TH821 and TH822 (CCP5), and TH895 and TH896 (CCP6) and assembled with EcoRI-digested vector with Gibson assembly. A catalytic dead version of CCP1 mutant (HS/EQ or H880S/E883Q) [6] was generated by site-directed mutagenesis using PrimeStar MAX DNA polymerase (Takara Bio, Kusatsu, Shiga, Japan) with a primer TH744. To construct EGFP-C1-mTTL, mouse TTL was amplified with oligonucleotide primers oMG70 and oMG71 and assembled with pEGFP-C1 that had been linearized by oMG68 and oMG69. Lipofectamine 2000 (Invitrogen, Cat# 11668-019, Waltham, MA, USA) was used for transient expression of EGFP-tagged CCPs or mTTL in tissue culture cells, and cells were lysed or fixed 24 h after transfection.

GST-CTT(Y), GST-CTT(∆Y) constructs were described previously [30]. GST-CTT (∆C2) and (∆C3) were generated in a similar manner. pGEX-KGT vector was PCR-amplified with phosphorylated primers TH511 and T514 (∆C2) or TH511 and TH515 (∆C3) and ligated with T4 DNA ligase (New England Biolabs, Ipswich, MA, USA, Cat# M0202S).

### 2.4. Quantseq

Total RNA prepared from HeLa or *TTLΔ* cell lines was submitted to the University of Michigan Advanced Genomics core for QuantSeq 3′ mRNA sequencing (https://doi.org/10.1038/nmeth.f.376, accessed on 1 December 2022). FASTQ files from the 100 bp sequencing run were first processed with the umi2index script supplied by Lexogen to strip the UMI from the reads. Cutadapt v1.15 was used to remove adapter resulting in 28, 26, 26, and 29 million reads for downstream analysis for samples AP-1, 2, 3, and 4, respectively. STAR v2.7.2b (https://doi.org/10.1093/bioinformatics/bts635, accessed on 1 December 2022) was used to align reads to the GRCh38 reference (command provided below). The Lexogen provided, collapse_UMI_bam script was then used to remove PCR duplicates.

Features were counted using featureCounts from the Rsubread Bioconductor package (https://doi.org/10.1093/nar/gkz114, accessed on 1 December 2022) with respect to features defined by the GTF file supplied with the GRCh38 reference. Normalized counts per million were generated using edgeR (https://doi.org/10.1093/bioinformatics/btp616, accessed on 1 December 2022) and the trimmed mean of M-values (TMM) method. Values were log2 transformed.

STAR command line:

STAR –runThreadN 6 –runMode alignReads –genomeDir ~/Homo_sapiens/NCBI/GRCh38/Sequence/STARIndex –outFilterType BySJout –outFilterMultimapNmax 20 –alignSJoverhangMin 8 –alignSJDBoverhangMin 1 –outFilterMismatchNmax 999 –outFilterMismatchNoverLmax 0.6 –alignIntronMin 20 –alignIntronMax 1,000,000 –alignMatesGapMax 1,000,000 –outSAMattributes NH HI NM MD –outSAMtype BAM SortedByCoordinate –limitBAMsortRAM 40000000000 –genomeLoad LoadAndKeep –readFilesCommand zcat –outFileNamePrefix 1588-AP-1__ –readFilesIn 1588-AP-1_trimmed.fastq.gz.

### 2.5. Immunoblot Analysis

Tissue culture cells: To validate the specificity of Y, ∆Y, and ∆C2 antibodies, GST-CTT (Y, ∆Y, ∆C2, and ∆C3) proteins were produced in and purified from bacteria as described previously [30]. To analyze whole cell lysates, cells were lysed with lysis buffer (6 mM Na_2_HPO_4_, 4 mM NaH_2_PO_4_, 2 mM EDTA, 150 mM NaCl, 1% NP40, and protease inhibitors) with a brief sonication followed by clarification. Protein concentration was measured by Bradford protein assay (Bio-Rad, Hercules, CA, USA, Cat# 5000006) using BSA as a standard. GST-CTT proteins (200 ng/lane), whole cell lysates (15 µg protein/lane except for CCP1 detection at 50 µg/lane), and purified bovine brain tubulin (200 µg/lane) were separated on 10% acrylamide gels and transferred onto nitrocellulose membranes. Blocking was performed with 5% skim milk solution in PBS supplemented with 0.05% tween 20 (PBST) at room temperature for 1 h, followed by incubation with primary (4 °C overnight) and secondary antibodies (room temperature for 1 h). Membranes were imaged with Azure c600 imager (Azure Biosystems, Dublin, CA, USA), and bands were quantified using Fiji (imageJ2 version 2.9.0/1.53t).

Mouse retina: A single P15 mouse retina was lysed in 250 µL of 2% SDS and 1× cOmpleteTM Protease (11836170001; Millipore Sigma; Bio-Rad, Hercules, CA, USA) diluted in 1X PBS and sonicated 3X for 5 s at 20% before clearing precipitates at 21,000× *g* for 30 min at room temperature. SDS-PAGE using AnykD Criterion TGX Precast Midi Protein Gels (5671124, Bio-Rad) was followed by transfer at 66 mV for 120 min onto Immun-Blot Low Fluorescence PVDF Membrane (1620264, Bio-Rad). Membranes were blocked using Intercept Blocking Buffer (927-70003; LI-COR Biosciences, Omaha, NE, USA). Primary antibodies were diluted in 50%/50% of Intercept/PBST and incubated overnight rotating at 4 °C. The next day, membranes were rinsed 3 times with PBST before incubating in the corresponding secondary donkey antibodies conjugated with Alexa Fluor 680 or 800 (LI-CORBiosciences) in 50%/50%/0.02% of Intercept/PBST/SDS for 2 h at 4 °C. Bands were visualized and quantified using the Odyssey CLx infrared imaging system (LiCor Bioscience).

Other antibodies used are as follows: GST antibody (Nacalai, San Diego, CA, USA, Cat# 04435-26), 1:1000; rat monoclonal anti-tyrosinated a-tubulin YL1/2 (Accurate Chemical and Scientific, Carle Place, NY, USA, Cat# YSRTMCA77G), 1:3000; rabbit monoclonal anti-detyrosinated α-tubulin RM444 (RevMAb Biosciences, San Francisco, CA, USA, Cat# 31-1335-00), 0.1 µg/mL; rabbit monoclonal anti-∆C2 α-tubulin RM447 (RevMAb Biosciences, Cat# 31-1339-00), 0.5 µg/mL; mouse monoclonal anti-α-tubulin DM1A (Millipore Sigma, Cat# 05-829), 1:3000; mouse monoclonal anti-GAPDH G-9 (Santa Cruz, Dallas, TX, USA, Cat# sc-365062), 1:2000; anti-TTL (Proteintech, Munich, Germany, Cat# 13618-1-AP), 1:1000; anti-CCP1 (Bethyl Laboratories, Montgomery, TX, USA, Rosemont, IL, USA, Cat# A305-295A), 1:1000; anti-GFP (Invitrogen, Cat# A6455) 1:2000; anti-rabbit Alexa Fluor 488 (Thermo Fisher, Waltham, MA, USA, Cat# A-11034), anti-mouse Alexa Fluor 488 (Thermo Fisher, Cat# A-21202), anti-rat Alexa Fluor 680 (Thermo Fisher, Cat# A-21096), 1:10,000; anti-mouse Alexa Fluor 700 (Thermo Fisher, Cat# A-21036), 1:10,000; anti-rabbit Alexa Fluor 700 (Thermo Fisher, Cat# A-21038), 1:10,000; anti-mouse DyLight 800 (Thermo Fisher Cat# SA5-10176), 1:10,000; anti-rabbit IRDye 800CW (LI-COR, Cat# 926-32211), 1:10,000.

### 2.6. Establishment of TTLΔ and TTLΔ CCP1Δ HeLa Cell Lines

CRISPR-Cas9 mediated gene knockout for TTL and CCP1 was performed according to the method developed by [31]. A single guide RNA (sgRNA) specific to TTL (CAGTTGGTGAATTACTACA) was cloned into pLenti-sgRNA that encodes puromycin resistance (Addgene, Watertown, MA, USA, #71409) [32,33] and introduced into HeLa cell line expressing inducible Cas9 (cTT20.11 cell line obtained from the Cheeseman lab) by lentiviral transduction. Cells were selected by 0.5 µg/mL puromycin for 7 days followed by Cas-9 induction with 1 µg/mL doxycycline for 7 days. Single cells were isolated by a cell sorter and expanded. Each clone was analyzed with immunoblot, and three clones exhibiting complete loss of TTL were identified. INDEL sequences were determined using Guide-it Indel Identification Kit (Takara Bio., Cat# 631444). To generate the *TTLΔ CCP1Δ* line, an sgRNA specific to CCP1 (GTTATTCCTGTGACTGGTCC) was cloned into a lentiviral vector pKM808 (Addgene, #134181) that encodes blasticidin resistance [31]. The guide RNA construct was introduced into the *TTLΔ* cells, and cells were selected with 2 µg/mL of blasticidin.

### 2.7. Targeted siRNA Screen against CCPs

To screen for CCPs that generate ΔC2-α-tubulin in HeLa cells lacking TTL, we used ON-TARGETplus siRNAs (SMARTPools) targeting CCP1-6 (CCP1: L-014059-00-0010; CCP2: L-012937-00-0010; CCP3: L-025456-01-0010; CCP4: L-017061-02-0010; CCP5: L-009468-00-0010; CCP6: L-014994-02-0010; Horizon Discovery, Waterbeach, UK). HeLa *TTLΔ* cells were plated into a 24-well dish and transfected 24 h later with 100 pmol of siRNA and HiPerFect (Qiagen, Hilden, DE, USA) according to the manufacturer’s instructions. Twenty-four hours later, cells were trypsinized and re-plated into a 6-well dish and transfected again with 100 pmol of CCP siRNAs and HiPerFect. For immunofluorescence, 12 mm acid-washed coverslips were placed into the wells, and cells were processed 48 h after transfection as described below. For immunoblotting, cells were lysed 48 h after transfection.

### 2.8. Immunofluorescence

Tissue culture cells: Immunofluorescence staining was performed as described before [29]. For taxol treatment, cells were cultured in the presence of 10 µM taxol for 12 h prior to the fixation. Antibodies used were rabbit monoclonal anti-detyrosinated α-tubulin RM444 (RevMAb Biosciences, Cat# 31-1335-00), 1 µg/mL; rabbit monoclonal anti-∆C2 α-tubulin RM447 (RevMAb Biosciences, Cat# 31-1339-00), 1 µg/mL; anti-α-tubulin DM1A conjugated with FITC (Sigma-Aldrich, St. Louis, MO, USA, Cat# F2168), 1:500; mouse monoclonal anti-α-tubulin DM1A (Millipore Sigma, Cat# 05-829), 1:1000; goat anti-mouse Alexa Fluor 647 (Thermo Fisher, Cat# A-32728), 1:2000; goat anti-rabbit Alexa Fluor 594 (Thermo Fisher, Cat# A-11012), 1:2000. DNA was counterstained with 5 mg/mL of Hoechst. Coverslips were mounted with ProLong Diamond (Thermo Fisher). Images were obtained with a DeltaVision microscope equipped with an Olympus Plan Apo N 60x/1.42 oil immersion lens. Images were deconvolved, and single optical sections were presented.

Fresh mouse retinas: Immunostaining protocol originally described in [34] with slight modifications. Fresh retinas were dissected in Supplemented Mouse Ringers, at pH 7.4 and ~313−320 mOsM [130 mM NaCl, 3.6 mM KCl, 2.4 mM MgCl_2_, 1.2 mM CaCl_2_, 10 mM HEPES, 0.02 mM EDTA, 20 mM Bicarbonate, 10 mM D-glucose, 1X vitamins, 0.5 mM Na(L)-glutamate, 3 mM Na_2_Succinate]. Fresh retinas were blocked in 10% normal donkey serum, 0.3% saponin, 1× cOmpleteTM Protease diluted in Supplemented Mouse Ringers for 1 h at 4 °C. Primary antibodies were diluted in blocking buffer and incubated for 2 h at 4 °C. Primary antibodies used are rabbit monoclonal anti-∆C2 α-tubulin RM447, 0.1 µg/mL, and mouse monoclonal anti-centrin1 (Millipore Sigma, Cat#04-1642), 1:1000. Retinas were rinsed and fixed in 4% paraformaldehyde for 30 min at room temperature. Retinas were rinsed and incubated for 2 h at 4 °C with donkey anti-mouse Alexa Fluor 488 (Fisher Scientific, A-21202), 1:2000; and donkey anti-rabbit Alexa Fluor 647 (Fisher Scientific, Cat# A-31573), 1:2000. Fixed stained retinas were then embedded in 4% agarose (BP160-500, Thermo Fisher Scientific) and cut into 200 µm thick sagittal sections with the vibratome. Sections were mounted with 1.5 mm coverslips (72204-10, EMS) using Prolong Gold (P36980, Thermo Fisher Scientific).

Isolated mouse outer segments: Immunostaining protocol originally described in [35]. To prepare isolated outer segments, retinas were dissected and collected in a microcentrifuge tube containing 125 µL of Mouse Ringers per retina. The retinas were vortexed on HIGH for 1 min and large debris pelleted using a benchtop spinner for 5 s. The supernatant containing isolated outer segments was then plated onto 13 mm poly-L-lysine glass coverslips (354085; Corning, Glendale, AZ, USA) before fixation in 4% paraformaldehyde in PBS at room temperature for 5 min. Plated outer segments were then rinsed with PBS and permeabilized for 5 min by incubating in 0.02% SDS diluted in PBS. Coverslips were rinsed, and primary antibodies were diluted in 2% donkey serum (Thermo Fisher Scientific, Cat# NC0629457) and 0.5% Triton X-100 in PBS and incubated for 2 h at room temperature. Primary antibodies used are rabbit monoclonal anti-∆C2 α-tubulin RM447, 0.1 µg/mL; rabbit monoclonal anti-detyrosinated α-tubulin RM444, 0.1 µg/mL; mouse monoclonal anti-α-tubulin DM1A (Sigma-Aldrich, Cat# T9026), 1:2000; mouse monoclonal anti-polyglutamylation GT335 (AdipoGen, San Diego, CA, USA, Cat# AG-20B-0020), 1:2000. Coverslips were rinsed and incubated with donkey secondary antibodies listed above for 1 h at room temperature before rinsing and mounting using Prolong Gold.

### 2.9. Image Analysis

Images from mouse retinal sections or isolated outer segments were acquired using a Zeiss Observer 7 inverted microscope equipped with a 63× oil-immersion objective (1.40 NA), LSM 800 confocal scanhead outfitted with an Airyscan super resolution detector controlled by Zen 5.0 software (Carl Zeiss Microscopy, Oberkochen, Germany). Manipulation of images was limited to adjusting the brightness level, image size, rotation, and cropping using FIJI (ImageJ, https://imagej.net/Fiji, accessed on 1 September 2017). All phenotypes were measured using images taken from at least 3 independent retinas.

Measuring fluorescence intensity of axoneme in isolated outer segments: For each antibody condition, the axoneme was stained with either α-tubulin or GT335 antibodies and outer segment membranes co-stained with WGA lectin conjugated to Alexa594 (W11262, Thermo Fisher Scientific). For each mouse, 10–15 isolated outer segments were imaged with an Airyscan detector on an LSM 800 microscope (Carl Zeiss Microscopy). Resolution was standardized to 0.04 µm/pixel by setting the axial zoom to 3X, and 0.35 µm Z-stacks were taken to collect the entire thickness of the outer segment, generally 0.9–1.3 µm in depth. A pre-determined laser power was set to the ccp1^+/+^ levels for each antibody condition. Using FIJI software, images were stacked, and a 15-pixel segmented line was drawn along the axoneme to acquire the fluorescence intensity measurements for each channel. The intensity measurements of each antibody were then aligned to the start of the connecting cilium, based on predetermined WGA parameters, and plotted using Prism 9 software.

## 3. Results

### 3.1. HeLa Cells Lacking TTL Contain High Levels of ΔC2-α-Tubulin

We began by examining the levels of ΔY-α-tubulin and ΔC2-α-tubulin in a panel of commonly used cell lines with PTM-specific rabbit monoclonal antibodies that we generated in collaboration with RevMAb Biosciences (clones RM444 and RM447 for ∆Y- and ΔC2-α-tubulin, respectively; Appendix A). By immunoblotting, the levels of ΔY-α-tubulin were found to be low in HeLa, DLD-1, HEK293T, U2OS, and CHL-1 cells when compared to bovine brain tubulin (Figure 1A). ΔC2-α-tubulin was undetectable in all of these cell lines under the conditions of our experiment. To determine if the enzymatic machinery that produces ΔC2-α-tubulin is present in HeLa cells, we used an inducible CRISPR/Cas9 system [36] to edit the TTL open reading frame (ORF) since deletion of this enzyme leads to widespread production of ΔC2-α-tubulin in mice [20]. We screened for potential *TTLΔ* clones by immunoblotting with ΔY-α-tubulin antibodies, and sequencing of one *TTLΔ* clonal isolate revealed three distinct insertions/deletions (INDELs; Figure 1B). Each INDEL presumably corresponds to one *TTL* allele, consistent with the literature, suggesting that various HeLa strains are hypertriploid [37,38]. All three INDELs produce frameshift mutations that result in early stop codons, and we confirmed that no TTL protein is present in cell lysates of this *TTLΔ* clone (Figure 1C).

We proceeded by staining the parental HeLa and *TTLΔ* cell lines with antibodies against α-tubulin (DM1A), ΔY-α-tubulin, and ΔC2-α-tubulin. Consistent with immunoblotting (Figure 1A), ΔY-α-tubulin levels were very low in HeLa cells, although short segments of ΔY-MTs were present in the cytoplasm of interphase cells (Figure 1D). ΔC2-α-tubulin was undetectable (Figure 1D), but we note that our antibody stains puncta in the nuclei of a subset of interphase cells, which likely results from cross-reactivity with a protein that is not tubulin. In contrast to the parental cell line, *TTLΔ* cells harbored very high levels of both ΔY-α-tubulin and ΔC2-α-tubulin (Figure 1D), a result that was recapitulated by immunoblotting of lysates prepared from these cell lines (Figure 1E). Levels of ΔY-α-tubulin and ΔC2-α-tubulin were reduced following re-introduction of mouse TTL into *TTLΔ* cells, indicating that high levels of ΔY-α-tubulin and ΔC2-α-tubulin are a specific consequence of TTL deletion (Appendix A). We obtained two additional *TTLΔ* clones and confirmed that the results were consistent among the three cell lines (Appendix A). By the reduction of Y-tubulin on immunoblots, we estimate that roughly half of the α-tubulin present in *TTLΔ* cells is either detyrosinated or in the ΔC2 form.

### 3.2. C2-α-Tubulin Levels Do Not Increase When Microtubules Are Artificially Stabilized

Several explanations could account for the increase in ΔC2-α-tubulin that is present in cells that lack TTL. The enzyme responsible for this PTM could be present but maintained in an inactive state, e.g., through an inhibitory mechanism(s). Alternatively, the peptidase could be induced at the transcriptional or post-transcriptional level upon removal of TTL. Since the six CCP genes are not differentially expressed in HeLa versus *TTLΔ* cells (Appendix A), this did not seem a likely possibility. Lastly, the catalytic activity of the peptidase may be too slow to keep pace with the tyrosination/detyrosination cycle. To investigate this, we treated cells with the microtubule stabilizing drug taxol, a perturbation well known to increase microtubule detyrosination [28]. This effect reflects the fact that VASH1-SVBP more efficiently detyrosinates microtubules, whereas TTL tyrosinates tubulin in its unpolymerized form [21,39,40]. Thus, we hypothesized that taxol treatment of cells would provide the ΔC2-α-tubulin peptidase with ample substrate (ΔY-α-tubulin) and time to generate ΔC2-α-tubulin.

We treated cell lines that harbor varying levels of ΔY-α-tubulin (HeLa, DLD-1, CHL-1, and *TTLΔ*) with 10 µM taxol for 12 h and then processed them for immunofluorescence (Figure 2A) or immunoblotting (Figure 2B). As expected, taxol treatment caused HeLa, DLD-1, and CHL-1 cells to express high levels of ΔY-α-tubulin and arrest in mitosis. Surprisingly, however, taxol treatment did not cause an increase in ΔC2-α-tubulin levels in these cell lines. The presence of both ΔY-α-tubulin and ΔC2-α-tubulin was only observed in taxol-treated *TTLΔ* cells (Figure 2A). To validate this result biochemically, we prepared cell lysates from DMSO- or taxol-treated HeLa, DLD-1, CHL-1, and *TTLΔ* cells and subjected them to immunoblotting analysis using antibodies against α-tubulin (DM1A), ΔY-α-tubulin, and ΔC2-α-tubulin. Mirroring observations made at the cellular level, we observed that taxol treatment increased the levels of ΔY-α-tubulin but not ΔC2-α-tubulin in HeLa, DLD-1, and CHL-1 cells. *TTLΔ* cells exhibited high levels of both ΔY-α-tubulin and ΔC2-α-tubulin regardless of drug treatment. Collectively, our results are inconsistent with the idea that ΔC2-α-tubulin levels are low because the enzyme responsible is too slow to keep pace with the tyrosination/detyrosination cycle.

### 3.3. Depletion or Gene Deletion of CCP1 Blunts the Production of ΔC2-α-Tubulin in TTLΔ Cells

To identify the peptidase that generates ΔC2-α-tubulin in *TTLΔ* cells, we first conducted a focused siRNA screen targeting the six CCPs (CCP1-6) encoded by the human genome. HeLa or *TTLΔ* cells were subjected to two rounds of siRNA transfections, fixed five days following the start of the experiment, and stained with antibodies against α-tubulin (DM1A) and ΔC2-α-tubulin. Depletion of CCP1 uniquely caused a reduction in the number of cells that were positive for ΔC2-α-tubulin (Figure 3A). By immunoblotting, we confirmed that CCP1 protein levels were reduced in siRNA-transfected cells, although depletion was incomplete. Immunoblotting also demonstrated that ΔC2-α-tubulin levels, but not ΔY-α-tubulin, dropped by 78% relative to control siRNA-transfected cells (Figure 3B). In parallel, we used QuantSeq to analyze the expression levels of CCP1-6 in HeLa and *TTLΔ* cells and found that CCP1 and CCP5 were the only two CCPs that are expressed at detectable levels (Appendix A). Since CCP5 is largely thought to remove the E residue from mono-glutamylated α-tubulin [6], it is reasonable that CCP1 is solely responsible for generating ΔC2-α-tubulin in *TTLΔ* HeLa cells.

To rule out the possibility that other enzymes generate ΔC2-α-tubulin in CCP1-depleted *TTLΔ* HeLa cells, we again used the CRISPR/Cas9 system [31] to disrupt the CCP1 gene in *TTLΔ* cells. We screened for potential *TTLΔ CCP1Δ* clones by immunoblotting with ΔC2-α-tubulin antibodies, and sequencing of one *CCP1Δ* clonal isolate revealed three distinct insertions/deletions (INDELs; Figure 3C). All three INDELs produce frameshift mutations that result in early stop codons in CCP1 isoform 1 (UniProt, Q9UPW5-1). For the isoform 2 that lacks internal 120-nt (UniProt, Q9UPW5-2; NCBI Reference Sequence, NM_015239.3), one of the INDELs produced a 16-nt deletion across intron 10 and exon 11, which most likely results in skipping of exon 11 (58-nt) and subsequent frameshift and an early stop codon. We confirmed that no CCP1 protein is present in cell lysates of this *TTLΔ CCP1Δ* clone (Figure 3D). *TTLΔ CCP1Δ* cells did not exhibit gross defects in cell proliferation (data not shown).

Immunostaining of the *TTLΔ CCP1Δ* cell line with antibodies against α-tubulin (DM1A), ΔY-α-tubulin, and ΔC2-α-tubulin revealed that ΔC2-α-tubulin, but not ΔY-α-tubulin, was absent in *TTLΔ CCP1Δ* during all phases of the cell cycle (Figure 3E and Appendix A). The staining pattern of ΔY-α-tubulin in *TTLΔ CCP1Δ* cells was not qualitatively different from *TTLΔ* single knockout cells. These observations are mirrored in immunoblots of lysates prepared from HeLa, *TTLΔ*, and *TTLΔ CCP1Δ* cells: ΔC2-α-tubulin was not detected in *TTLΔ CCP1Δ* lysates, whereas ΔY-α-tubulin levels were unchanged relative to lysates prepared from *TTLΔ* cells. Like *TTLΔ* cells, Y-α-tubulin levels are decreased by ~50% in *TTLΔ CCP1Δ* cells (Figure 3D). We conclude that CCP1 is the sole peptidase that generates ΔC2-α-tubulin in *TTLΔ* cells and that α-tubulin in the *TTLΔ CCP1Δ* cell line is either tyrosinated or detyrosinated.

### 3.4. CCP1 Requires ΔY-α-Tubulin as a Substrate to Generate ΔC2-α-Tubulin

While levels of total Y- and ∆Y-α-tubulin were not affected in *TTL∆ CCP1∆* cells compared to *TTL∆* cells (Figure 3D), ∆C2-tubulin in the *TTL∆* line seemed to represent only a small portion of total α-tubulin. One possible explanation is that removal of residues from the α-CTT occurs in a stepwise fashion, i.e., ∆Y-α-tubulin is generated first, followed by production of ∆C2-tubulin. If this were the case, CCP1 overexpression should generate ΔC2-α-tubulin more efficiently in cells that contain higher levels of ΔY-α-tubulin. To test this, we overexpressed EGFP-tagged CCP1 or a catalytically dead version of the enzyme (H880S/E883Q, hereafter HS/EQ, [6]) in HeLa, HEK293T, or CHL-1 cells; HeLa cells do not harbor significant levels of ΔY-α-tubulin, whereas ΔY-α-tubulin is readily detectable in HEK293T and CHL-1 cells (Figure 1A). HeLa and HEK293T cells were efficiently transfected with our constructs, and we could thus test the effect of CCP1 overexpression on ΔC2-α-tubulin levels in these cell lines by both immunostaining and immunoblotting. In HeLa cells, ΔC2-α-tubulin was undetectable in both immunoblots and immunostained cells regardless of whether they were transfected with wild-type EGFP-CCP1 or the HS/EQ CCP1 mutant. In contrast, overexpression of wild-type CCP1, but not the HS/EQ mutant, led to an increase in the amount of ΔC2-α-tubulin in HEK293T cells (Figure 4A,B and Appendix A). ΔC2-α-tubulin decorated perinuclear portions of microtubules, which presumably represent minus ends that are concentrated at the centrosome(s) (Figure 4B). Unfortunately, CHL-1 cells were not efficiently transfected, precluding a test of whether CCP1 overexpression can increase ΔC2-α-tubulin levels by immunoblotting. However, immunostaining of CHL-1 cells revealed that ΔC2-microtubules were abundant in interphase cells transfected with wild-type CCP1 but not the HS/EQ mutant (Figure 4B and Appendix A).

To conclusively demonstrate that CCP1 uses ΔY-microtubules as a substrate to generate ΔC2-α-tubulin, we transfected HeLa cells that inducibly express VASH1-SVBP with wild-type CCP1 or the HE/SQ mutant. VASH1-SVBP expression was induced by the addition of doxycycline for 16 h and then transfected with EGFP-CCP1 or EGFP-CCP1^HS/EQ^. Cells were processed for immunofluorescence or harvested for immunoblot analysis 24 h following transfection. These experiments clearly demonstrated that overexpression of wild-type CCP1, but not the HE/SQ mutant, caused an increase in the levels of ΔC2-α-tubulin only in cells that were induced to express VASH1-SVBP (Figure 5 and Appendix A). We conclude that proteolytic trimming of the α-CTT occurs in a stepwise manner: VASH1-SVBP first removes the C-terminal Y residue, producing ΔY-α-tubulin that is then used as a substrate for CCP1 to generate ΔC2-α-tubulin.

### 3.5. CCP1 Converts Only a Subset of Photoreceptor Axoneme Microtubules to ΔC2-α-Tubulin

To investigate whether CCP1 can generate ΔC2-α-tubulin in an in vivo context, we looked at photoreceptor cells in the mouse retina that contain a specialized light-sensitive primary cilium, called the outer segment. The outer segment contains all the structural features of a primary cilium, including a microtubule axoneme emanating from the basal body and extended transition zone referred to as the connecting cilium but has been modified to house hundreds of flattened membrane vesicles for efficient light capture. A mouse model with mutation in CCP1 was shown to slowly lose photoreceptor cells during the first year of life in addition to rapid loss of cerebellar Purkinje cells [41]. Due to the ongoing photoreceptor degeneration present in the *ccp1* knockout mice [42,43], we analyzed axoneme tubulin PTM modifications at postnatal day 15, a timepoint when the ciliary outer segment is formed but ongoing degeneration has not begun. Fresh mouse retinal cross-sections were stained with antibodies against ΔC2-α-tubulin and centrin-1 to label the connecting cilium and counterstained with Alexa-594-conjugated WGA that labels outer segment membranes (Figure 6A). Confocal imaging of the outer segment showed reduced levels of ΔC2-α-tubulin in the *ccp1^−/−^* mice compared to littermate controls. To quantify tubulin staining intensity, we isolated outer segments from *ccp1^−/−^* and *ccp1^+/+^* mice at P15, plated onto coverslips, and stained outer segments with WGA-594 and antibodies against α-tubulin (DM1A) and ΔC2-α-tubulin. Airyscan images were acquired of individual outer segments so that intensity measurements could be collected along the axoneme. Axonemal intensity measurements across outer segments were aligned to the beginning of the connecting cilium based on a predetermined threshold of WGA signal. While *ccp1^+/+^* mouse outer segments contain an even distribution of ΔC2-α-tubulin across the axoneme, the intensity of ΔC2-α-tubulin staining is reduced outside the connecting cilium in *ccp1^−/−^* outer segments (Figure 6B). In contrast, ΔY-α-tubulin staining of the photoreceptor axoneme did not change in isolated *ccp1^−/−^* outer segments (Appendix A). A ~20% reduction in ΔC2-α-tubulin was also observed by immunoblot analysis from retina lysates of *ccp1^+/+^* and *ccp1^−/−^* mice (Figure 6C). Together, our data suggest that CCP1 is used to generate axonemal ΔC2-α-tubulin in photoreceptor outer segments outside of the connecting cilium. In addition, our data indicate that a second peptidase is capable of generating ΔC2-α-tubulin within the connecting cilium, which presumably comprises a large fraction of ΔC2-α-tubulin in retinal tissue.

### 3.6. CCP6 Is a Candidate for a Second Peptidase That Generates ΔC2-α-Tubulin

To identify other CCPs that can generate ΔC2-α-tubulin, we cloned full-length open-reading frames of CCP1-6 into pEGFP-C1 and assessed the ability of these constructs to produce ΔC2-α-tubulin when transfected into HeLa *TTLΔ CCP1Δ* cells. This cell line is ideal to screen for enzymes that generate ΔC2-α-tubulin because this PTM is absent when CCP1 is deleted in HeLa cells and the precursor to ΔC2-α-tubulin, ΔY-α-tubulin, is abundant (Figure 3E). Immunostaining of transfected cells with antibodies against ΔC2-α-tubulin showed that CCP1 and CCP6 were the most effective at generating ΔC2-α-tubulin in *TTLΔ CCP1Δ* cells and that CCP3 produced a small amount of ΔC2-α-tubulin (Figure 7).

## 4. Discussion

In contrast to many tubulin PTMs, such as poly-glutamylation, poly-glycylation, and acetylation, the function(s) of ΔC2-α-tubulin is unknown. ΔC2-α-tubulin is enriched in highly stable microtubules, such as those in neurons, the primary cilium, and centrosomes [5], but whether ΔC2-α-tubulin emerges as a consequence of or promotes microtubule stability has not been determined. A major reason for this knowledge gap is that ΔC2-α-tubulin-generating enzymes—the CCPs—have dual functions in post-translationally modifying microtubules, complicating analysis. CCPs remove glutamate residues that are appended onto the C-terminal tail of α-tubulin, i.e., poly-glutamate chains, as well as those that are encoded at the -2 and -3 positions by the α-tubulin genes [6,23]. Most work on CCPs has focused on their role in regulating poly-glutamate chains, as this activity is required to prevent the degeneration of multiple neuronal sub-types including Purkinje cell [27], motor [26], and retinal neurons [44]. A second reason is that we do not yet know the enzymes that generate ΔC2-α-tubulin in a physiological context. CCP1, a widely expressed CCP1 (Appendix A and [23]), has been shown to generate ΔC2-α-tubulin when overexpressed in cultured cells [6,23] and to be required for the formation of ΔC2-α-tubulin in HEK293T cells [24]. However, only skeletal muscle is depleted of ΔC2-α-tubulin in tissues of *ccp1^−/−^* mice [6], indicating that other enzymes cooperate with CCP1 to generate ΔC2-α-tubulin in differentiated cells.

In this work, we used knowledge that deletion of TTL increases the levels of ΔC2-α-tubulin [20] to screen for CCPs that generate ΔC2-α-tubulin in HeLa cells. Previous work to study CCPs employed HEK293T cells, which contain only low levels of ΔY-α-tubulin (Figure 1A). In contrast, ΔY-α-tubulin and ΔC2-α-tubulin comprise ~half of the α-tubulin pool in *TTLΔ* cells (Figure 1C). Since ΔY-α-tubulin is a prerequisite for the formation of ΔC2-α-tubulin (this work), *TTLΔ* cells are a robust model system to study the enzymes that generate the ΔC2, and presumably ΔC3 [4], form of α-tubulin. A second advantage of using *TTLΔ* cells to study the biogenesis of ΔC2-α-tubulin is that the ΔC2-α-tubulin-generating reaction appears to be gated by TTL through a mechanism that is unclear. Specifically, treatment of TTL+ cells with taxol for 24 h increases the levels of ΔY-α-tubulin but not ΔC2-α-tubulin. The reason for this is not clear. CCP1 may not be able to compete with other α-tubulin C-terminal tail-binding proteins or may be kept inactive through an unknown regulatory mechanism(s). Future work is needed to test these possibilities.

*TTLΔ* cells did not exhibit gross proliferation defects or abnormal progression through cell division (data not shown), which is consistent with the idea that *TTLΔ* cells adapt to heightened levels of ΔY-α-tubulin [45] and the ability of *ttl^−/−^* mice to undergo normal embryonic development [20]. In HeLa cells lacking TTL, CCP1 is the sole carboxypeptidase that generates ΔC2-α-tubulin, which we demonstrated unambiguously by showing that ΔC2-α-tubulin is absent in HeLa cells lacking both TTL and CCP1. Like *TTLΔ* cells, cells lacking both TTL and CCP1 did not exhibit defects in cell proliferation, suggesting that ΔC2-α-tubulin is not required for viability of HeLa cells. Of the five remaining CCPs, only CCP5 is expressed at detectable levels in HeLa cells. CCP5 is unique among the CCPs in that it (1) removes the E residue from mono-glutamylated α-tubulin and (2) does not generate ΔC2-α-tubulin when overexpressed [6]. It is therefore reasonable that ΔC2-α-tubulin cannot be produced in HeLa cells lacking CCP1, and an interesting direction for future work will be to survey the levels of ΔC2-α-tubulin in various tissues from *ttl^−/−^ ccp1^−/−^* mice.

Our work in photoreceptors is consistent with the notion that other CCPs can generate ΔC2-α-tubulin in differentiated tissues [6]. Indeed, the levels of ΔC2-α-tubulin are only decreased by ~20% in retinal extracts prepared from *ccp1^−/−^* mice (Figure 6C). Unexpectedly, however, we observed that ΔC2-α-tubulin is specifically depleted from a region of the photoreceptor cilium of *ccp1^−/−^* mice that is distal to the connecting cilium (Figure 6B), whereas the fluorescence intensity of ΔC2-α-tubulin in the connecting cilium is comparable to wild-type photoreceptors. This finding suggests that the activities of at least two CCPs are directed towards different regions of the outer segment: CCP1 targets the outer segment distal to the connecting cilium, whereas another CCP(s) generates ΔC2-α-tubulin within the connecting cilium. Previous work showed that CCP1-4 and CCP6 can generate ΔC2-α-tubulin when overexpressed in HEK293T cells [6,23]. However, the low levels of ΔY-α-tubulin in HEK293T cells, coupled with the use of truncated CCP2 and CCP3 constructs in earlier work [23], motivated us to reinspect the activities of the CCPs in *TTLΔ CCP1Δ* cells. Although we cannot rule out the possibility that co-factors, e.g., regulatory binding partners, are required for CCPs to be active in HeLa cells, our data strongly suggest that CCP6 and CCP1 are best positioned to generate ΔC2-α-tubulin in mammalian cells. Interestingly, CCP6 is not expressed at significant levels in rod photoreceptors (Appendix A) prior to P15, the time at which we processed our specimens for analysis. Therefore, it is currently unclear which CCP is responsible for generating ΔC2-α-tubulin in the outer segment, and this outstanding issue will be a focus for future work.

## Figures and Tables

**Figure 1 biomolecules-13-00357-f001:**
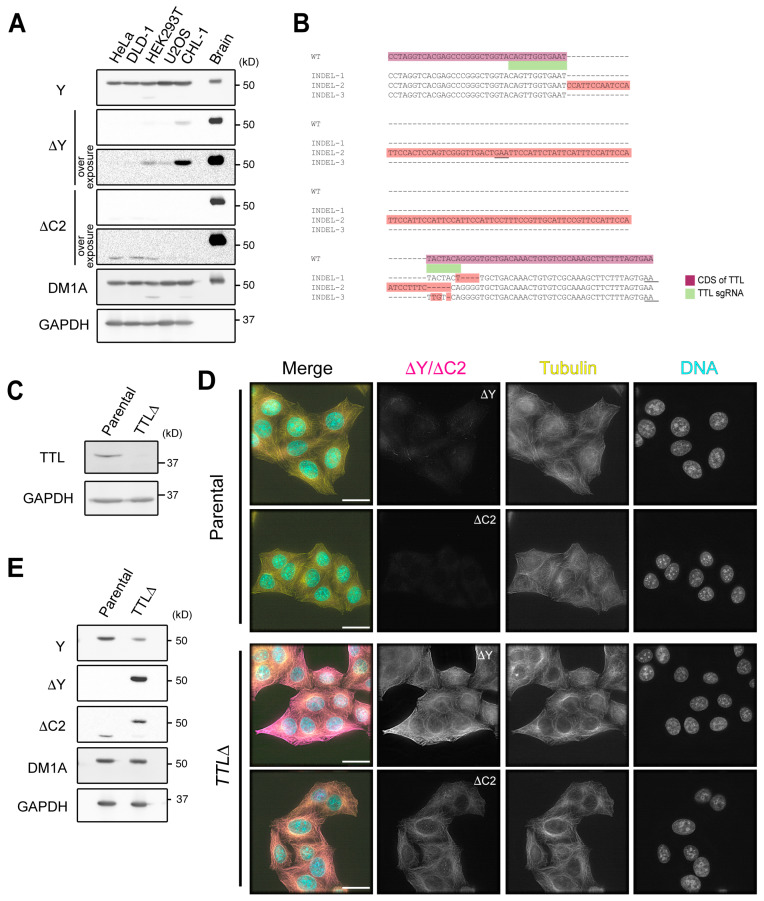
ΔC2-α-tubulin levels increase in *TTLΔ* HeLa cells. (**A**) Immunoblotting of whole cell lysates prepared from various tissue culture cell lines. ΔC2-α-tubulin was virtually undetectable. (**B**) Detection of CRISPR/Cas9-induced INDELs in the *TTL∆* line (clone 1). Three independent INDELs, all of which resulted in early stop codons (underlined), were identified. (**C**) Immunoblotting of whole cell lysates prepared from the parental HeLa and *TTL∆* lines. (**D**,**E**) Detection of ∆Y- and ∆C2-α-tubulin in the *TTL∆* line by immunofluorescence (**D**) and immunoblotting in cell lysates (**E**). In merged images in (**D**), ∆Y- or ∆C2-α-tubulin is shown in magenta, total α-tubulin (DM1A staining) in yellow, and DNA (DAPI staining) in cyan. Scale bars in (**D**), 25 µm.

**Figure 2 biomolecules-13-00357-f002:**
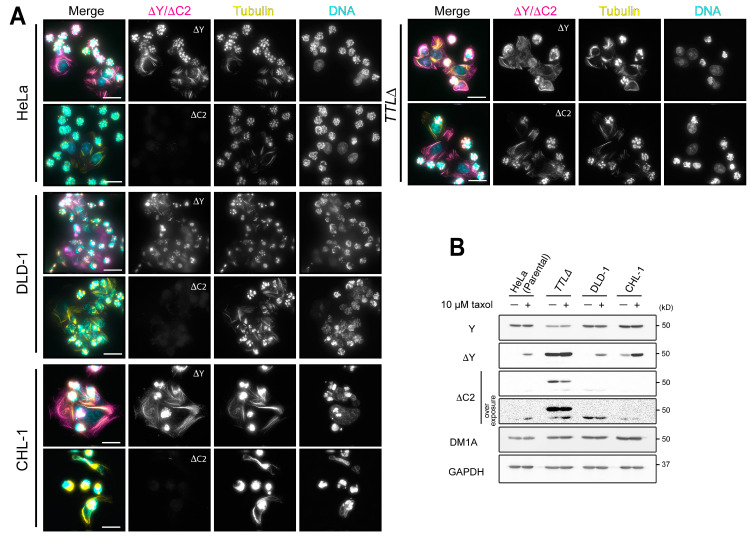
Taxol treatment increases the levels of ΔY-α-tubulin but not ΔC2-α-tubulin in cells. (**A**) Detection of ∆Y- and ∆C2-α-tubulin in taxol-treated HeLa, DLD-1, CHL-1, and *TTL∆* cells by immunofluorescence (**A**) and immunoblotting in whole cell lysates (**B**). In merged images in (**A**), ∆Y- or ∆C2-α-tubulin is shown in magenta, total α-tubulin (DM1A staining) in yellow, and DNA (DAPI staining) in cyan. Scale bars in (**A**), 25 µm.

**Figure 3 biomolecules-13-00357-f003:**
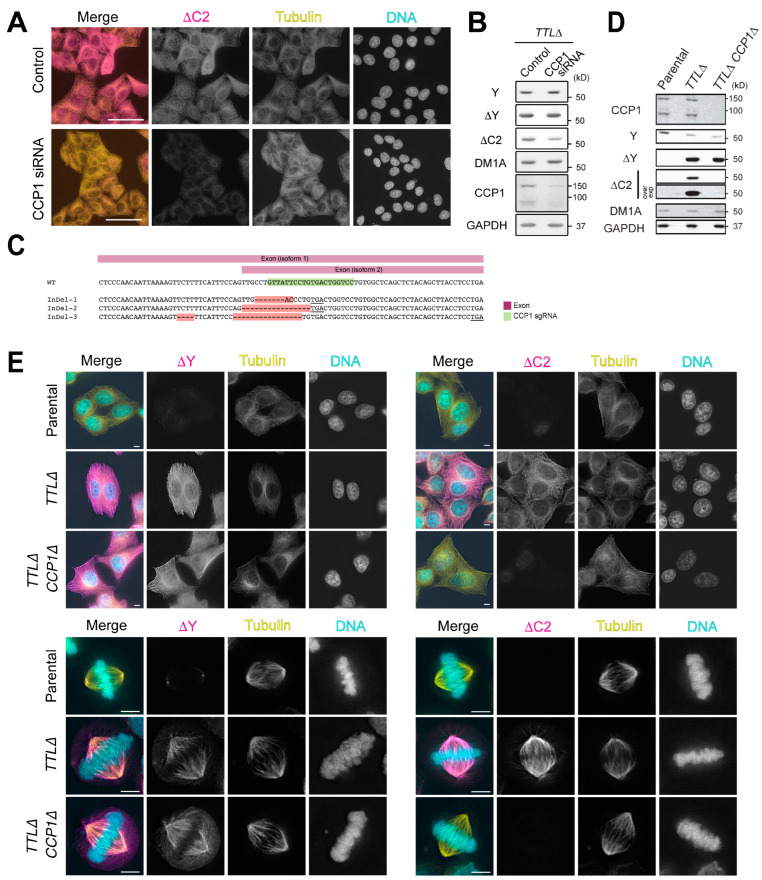
CCP1 is the sole peptidase that generates ΔC2-α-tubulin in *TTLΔ* cells. (**A**,**B**) siRNA-mediated knock down of CCP1 in *TTLΔ* cells, followed by detection of ΔC2-α-tubulin with immunofluorescence (**A**) and immunoblotting (**B**). In merged images in (**A**), ∆C2-α-tubulin is shown in magenta, total α-tubulin (DM1A staining) in yellow, and DNA (DAPI staining) in cyan. Scale bars in (**A**), 50 µm. (**C**) Detection of CRISPR/Cas9-induced INDELs in the CCP1 locus in *TTL∆ CCP1∆* line. Expected premature stop codons were indicated by underlines. (**D**,**E**) Detection of ∆C2-α-tubulin in the *TTL∆ CCP1∆* line by immunoblotting in cell lysates (**D**) and immunofluorescence (**E**). In merged images in (**E**), ∆Y- or ∆C2-α-tubulin is shown in magenta, total α-tubulin (DM1A staining) in yellow, and DNA (DAPI staining) in cyan. Scale bars in (**E**), 5 µm.

**Figure 4 biomolecules-13-00357-f004:**
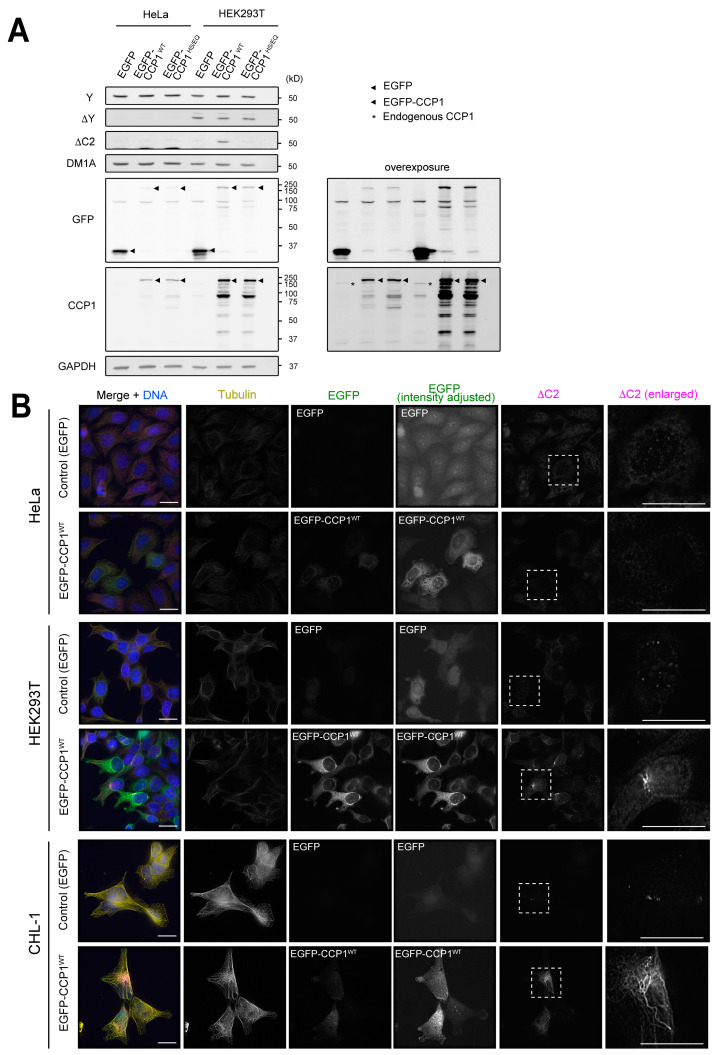
Overexpression of CCP1 generates ΔC2-α-tubulin more efficiently in cell lines harboring higher levels of ∆Y-α-tubulin. (**A**,**B**) Detection of ∆C2-α-tubulin in HeLa, HEK293T, or CHL-1 cells transiently expressing EGFP-CCP1 by immunoblotting of cell lysates (**A**) and immunofluorescence (**B**). In merged images in (**B**), total α-tubulin (DM1A staining) is shown in yellow, EGFP in green, ∆C2-α-tubulin in magenta, and DNA (DAPI staining) in blue. Scale bars in (**B**), 20 µm.

**Figure 5 biomolecules-13-00357-f005:**
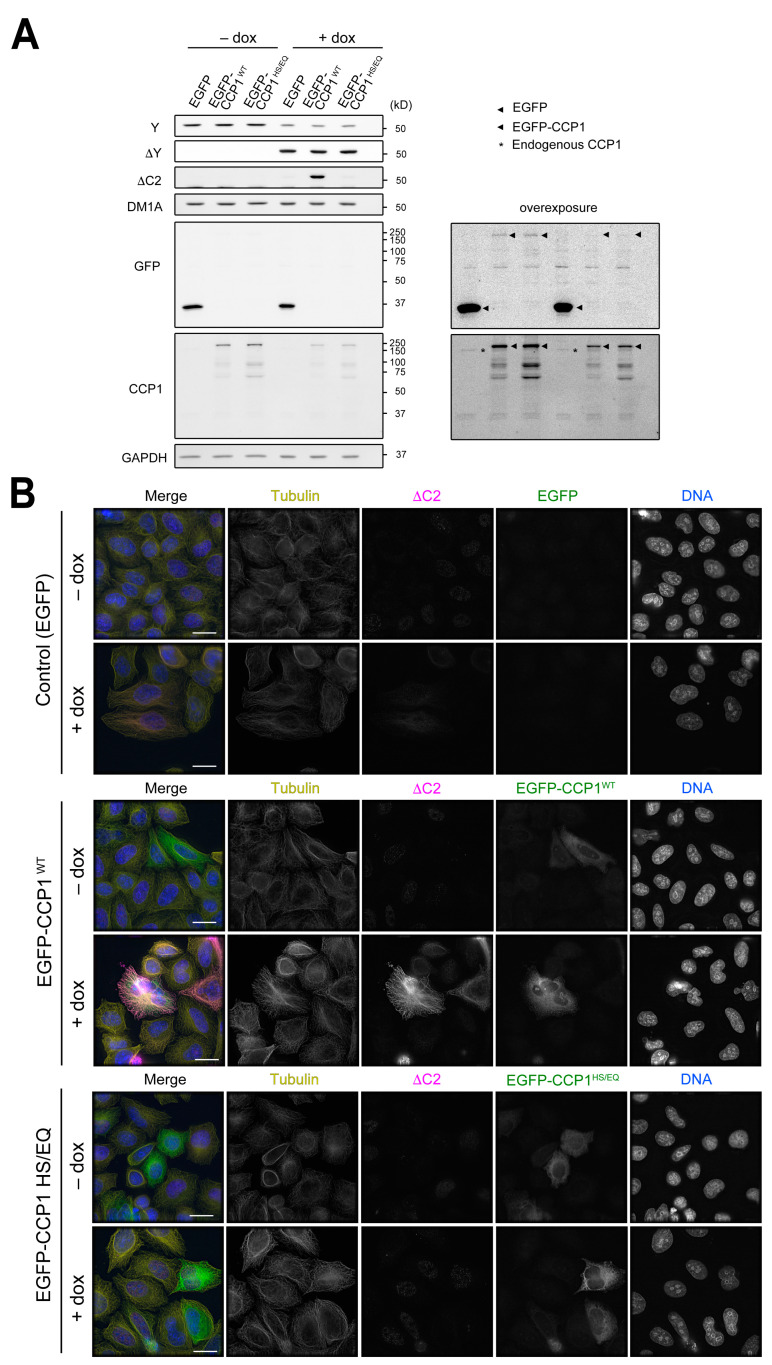
CCP1 uses ∆Y-α-tubulin as a substrate to generate ∆C2-α-tubulin in cells. (**A**,**B**) Detection of ∆C2-α-tubulin in HeLa cells expressing VASH1-SVBP in a doxycycline dependent manner by immunoblotting of cell lysates (**A**) and immunofluorescence (**B**). In merged images in (**B**), total α-tubulin (DM1A staining) is shown in yellow, ∆C2-α-tubulin in magenta, EGFP in green, and DNA (DAPI staining) in blue. Scale bars in (**B**), 20 µm.

**Figure 6 biomolecules-13-00357-f006:**
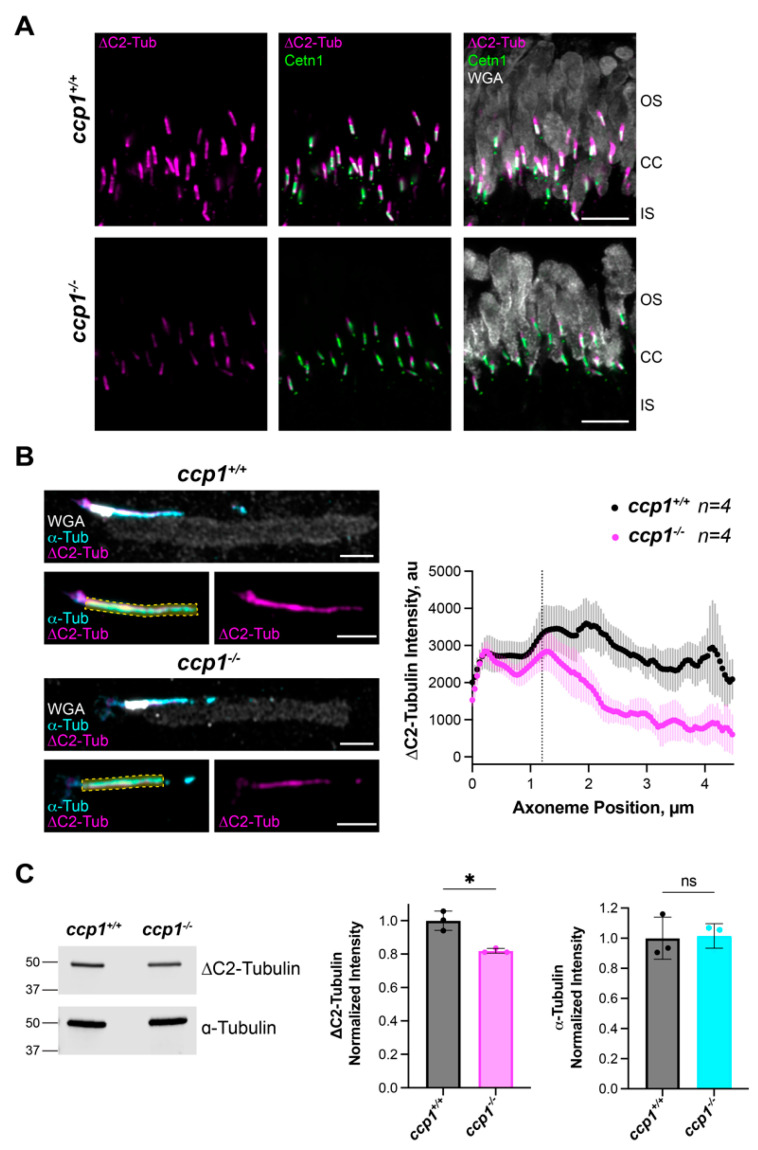
In *ccp1^−/−^* mouse retinas, ΔC2-α-tubulin staining is reduced distal to the connecting cilium in rod photoreceptor cells. (**A**) Representative images from P15 *ccp1^+/+^* and *ccp1^−/−^* retinal cross-sections stained with anti-centrin1 (green), WGA (grey), and anti-∆C2-α-tubulin (magenta) antibodies. Scale bars, 5 µm. Abbreviations: outer segment (OS), connecting cilium (CC), and inner segment (IS). (**B**) Representative Airyscan images of isolated outer segments from P15 *ccp1^+/+^* and *ccp1^−/−^* retinas stained with α-tubulin (cyan), WGA (gray), and anti-∆C2-α-tubulin (magenta) antibodies. Scale bar, 2 µm. Averaged intensity plots shown to right. (**C**) Representative Western blot showing anti-∆C2-α-tubulin and anti-α-tubulin bands from *ccp1^+/+^* and *ccp1^−/−^* retinal lysates. ∆C2-α-tubulin and α-tubulin levels at P15 were quantified for 3 separate mice and normalized to wild-type (*ccp1^+/+^*) levels. * *p* = 0.0438; ns, not significant *p* = 0.8528.

**Figure 7 biomolecules-13-00357-f007:**
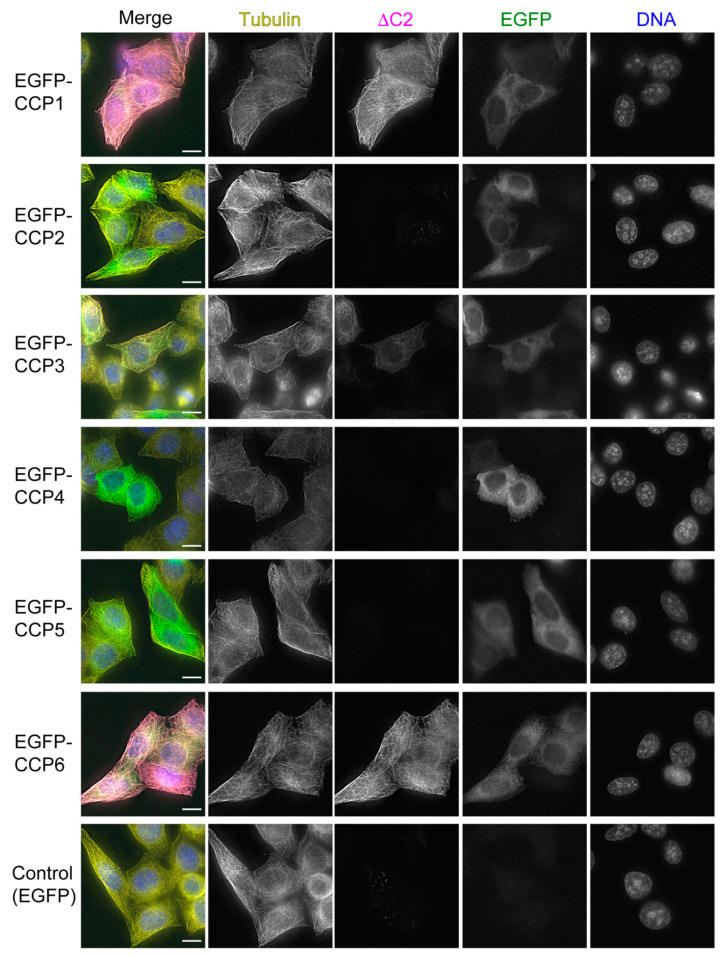
CCP6 is a candidate for a second peptidase that generates ΔC2-α-tubulin. Immunofluorescence of *TTL∆ CCP1∆* cells transiently expressing EGFP-tagged CCPs. In merged images, total α-tubulin (DM1A staining) is shown in yellow, ∆C2-α-tubulin in magenta, EGFP in green, and DNA (DAPI staining) in blue. Scale bars, 10 µm.

## Data Availability

Not applicable.

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
