# Peer review of "Mechanistic Analysis of CCP1 in Generating ΔC2 α-Tubulin in Mammalian Cells and Photoreceptor Neurons"

_biomolecules, 2023, doi:10.3390/biom13020357_

Round 1

Reviewer 1 Report

This study addresses the generation and regulation of DC2 tubulin in mammalian cells. Building on a previous paper , Ohi and colleagues determine that DC2 tubulin requires detyrosinated tubulin as a substrate. In fact, tubulin Tyr ligase-deprived cells accumulate detyrosinated tubulin, and the abundance of substrate likely enables the appearance of DC2 tubulin, which is otherwise undetectable in most cell lines. Using siRNA and overexpression, the authors show that cytoplasmic carboxypeptidase-1 (CCP1) is involved in the appearance of DC2 tubulin.

This paper is not bad at all, but it requires some experimentation to consolidate and substantiate the claims made. It is imperative that the authors are given enough time (1-2 months maybe) to address these points.

Perhaps the most important question, although this paper does not require to address it, is to what the physiological function of DC2 tubulin is. Speculation to this regard is kept to a minimum in the paper. I understand the authors do not want to get into details on this, but it is a crucially important point to make sense of the strong experimental effort the authors have made.

-          In general, immunofluorescence images are too small and could benefit from increased size. Particularly glaring in Fig. 3, but all are too small except Fig. 6, which is okay.

-          Several important controls are missing. In Fig. 1, rescue data in TTLD cells are necessary. Just overexpress TTL in TTLD cells to make sure DC2 tubulin disappears.

-          The authors argue that DC2 tubulin is undetectable in cell lines except HeLa because they contain a very small amount of detyrosinated tubulin, hence there is no substrate for CCP1 to cleave E -1. However, taxol-treated cell lines contain plenty of detyrosinated tubulin, yet levels of DC2 tubulin are unaffected. This needs to be addressed further. Do taxol-treated HeLa cells overexpressing CCP1 accumulate increased levels of DC2 tubulin? If not, how is that possible?

-          Is there anything noticeable about the localization of EGFP-CCP1? Are the authors sure the inclusion of the EGFP tag does not affect the activity and/or functionality and/or localization of CCP1?

-          Given that one central hypothesis of the study is that DC2 tubulin emanates from detyrosinated tubulin, it is important to reestablish that rod photoreceptor cells used in Fig.6 contain high levels of detyrosinated tubulin. While this is likely true, it needs to be shown here for consistency.

-           What are the levels of endogenous CCP1-6 in HeLa cells?

Reviewer 2 Report

The manuscript by Hotta et al. aims at describing the mechanisms that control the generation of detyrosinated and deglutamylated (∆2) tubulin in living cells. Even though the involvement of CCP1 has been proposed more than 10 years ago now from in vitro studies (see Berezkiuk et al 2012, JBC, 287:6503-17), nothing very discriminant on the roles played by cytosolic carboxypeptidases in generating ∆2 tubulin has been proposed yet. This study is therefore sound and it was clearly written, but my global impression is that some expectations of the reader could have been better addressed, without a final feeling of partial dissatisfaction.

Main concerns: 

- My first frustration is conceptual as the authors asked why ∆2-tubulin in non-neuronal cells can only be detected after the loss of TTL activity but never answered to this question. Actually this is the most interesting and relevant question of the study. They tried to address it with the Taxol experiments, but rapidly dropped this idea, which was no longer reused, even in the discussion. Also I think that among the hypotheses that are proposed to explain this peculiar behaviour, several options are missing. The authors restrict ∆2 tubulin generation to a problem of carboxypeptidase control of expression or activity but what they did explore is the availability of the substrate (detyrosinated tubulin), and did not consider the access to the C-terminal tail of alpha-tubulin (a possible competition with TTL? what happens between the soluble and the polymer compartments?). At the very least, the manuscript should be re-written to take this remark into account. 

- My second point is why did the authors wait for Figure 7 to show this key experiment of CCP over-expression and visualisation of ∆2-tubulin ?  Why did they conclude only about CCP1 (see the summary) and not also on CCP6 and CCP3? Of course this dataset has some limits due to non-physiological enzyme expression, but could it have been compared to some quantitative PCR data from cells or expressed in gradually decreasing amounts? I think the authors should address these questions and their paper should be remodelled in a reverse way, starting on a broad CCP family effects and then focusing on CCP1.

- At the beginning of § 3.4 the authors wrote: "While levels of total, Y- and ∆Y-alpha-tubulin were not affected in TTL∆ CCP1∆ cells 429 compared to TTL∆ cells (Figure 3D), ∆C2-tubulin in the TTL∆ line seems a small portion 430 of total alpha-tubulin. This suggests that removal of residues from the alpha-CTT occurs in a stepwise fashion..." I cannot see the logic between the first and the second sentences. A better approach to justify that generation of ∆2 tubulin is a two-step process is that CCPs cannot remove the terminal tyrosine, so detyrosination has to occur first...To be conclusive on this part, the correct approach would have been to prevent detyrosination (not to enhance it as shown here) by knocking down VASH/SVBP activity in a context of CCP1 overexpression. These remarks should be taken into account to reconsider the order of ideas and the logic of this paragraph.

Minor points:

If Figure 1D, the top right image is not a DNA signal but a copy of the ∆Y/∆2 signal.

The anti-alpha-tubulin antibody spells DM1A, not DM1alpha

Round 2

Reviewer 1 Report

The authors are to be commended for a stellar revision work. This study is excellent.

Author Response

Reviewer #1

COMMENT: The authors are to be commended for a stellar revision work. This study is excellent.

RESPONSE: We thank the reviewer for their enthusiasm regarding our manuscript.

Reviewer 2 Report

This revised version and the authors' response addressed all the points I raised, most of them being correctly answered or clarified.

However, regarding my point #2 there is still something that I don't feel comfortable with:

Excerpt of the authors' response:

We humbly disagree with this approach of organizing the manuscript for two major reasons. First, CCP6 and CCP3 and are not expressed in HeLa cells. This rules out the possibility that these peptidases are responsible for generating DeltaC2-a-tubulin in TTLDelta cells.

My feeling:

OK, I can understand this choice, but  in this case, why does the title convey a message that generalizes to the mechanisms of ∆C2 generation in mammalian cells ? To be consistent, the authors  should specifically focus on CCP1 in the title too.

and regarding point #3:

Authors' response:

We thank the reviewer for these comments. We have edited the manuscript text to clarify this point, as suggested by the reviewer. In response to the second point, it is not necessary to knock out VASH1 and/or SVBP in HeLa cells to prevent the accumulation of DC2-a-tubulin by CCP1 overexpression. In Fig 4, we show that overexpression of CCP1 does not cause the appearance of DC2-a-tubulin. DC2-a-tubulin is only produced by CCP1 if VASH1/SVBP is also over-expressed (Fig 5). We hope that this explanation clarifies this point.

My feeling:

OK. I agree with the author’s conclusions from HeLa data. A probable misleading came from the discrepancy with HEK data in which ∆C2 increased, due to the presence of basal ∆Y tubulin amounts. I suggest a clarification of this logic in the main text, maybe as an intermediate conclusion sentence by line 465 or so.  

Author Response

Reviewer #2

This revised version and the authors' response addressed all the points I raised, most of them being correctly answered or clarified.

However, regarding my point #2 there is still something that I don't feel comfortable with:

Excerpt of the authors' response:

We humbly disagree with this approach of organizing the manuscript for two major reasons. First, CCP6 and CCP3 and are not expressed in HeLa cells. This rules out the possibility that these peptidases are responsible for generating DeltaC2-a-tubulin in TTLDelta cells.

My feeling:

OK, I can understand this choice, but  in this case, why does the title convey a message that generalizes to the mechanisms of ∆C2 generation in mammalian cells ? To be consistent, the authors  should specifically focus on CCP1 in the title too.

We thank the reviewer for this comment. We have revised the title of the manuscript to “Mechanistic Analysis of CCP1 in Generating DC2-a-tubulin in Mammalian Cells and Photoreceptor Neurons”

and regarding point #3:

Authors' response:

We thank the reviewer for these comments. We have edited the manuscript text to clarify this point, as suggested by the reviewer. In response to the second point, it is not necessary to knock out VASH1 and/or SVBP in HeLa cells to prevent the accumulation of DC2-a-tubulin by CCP1 overexpression. In Fig 4, we show that overexpression of CCP1 does not cause the appearance of DC2-a-tubulin. DC2-a-tubulin is only produced by CCP1 if VASH1/SVBP is also over-expressed (Fig 5). We hope that this explanation clarifies this point.

My feeling:

  1. I agree with the author’s conclusions from HeLa data. A probable misleading came from the discrepancy with HEK data in which ∆C2 increased, due to the presence of basal ∆Y tubulin amounts. I suggest a clarification of this logic in the main text, maybe as an intermediate conclusion sentence by line 465 or so.

In the manuscript, we state (lines 441-448):

One possible explanation is that removal of residues from the a-CTT occurs in a stepwise fashion, i.e., ∆Y-a-tubulin is generated first, followed by production of ∆C2-a-tubulin. If this were the case, CCP1 overexpression should generate ∆C2-a-tubulin more efficiently in cells that contain higher levels of ∆Y-a-tubulin. To test this, we overexpressed EGFP-tagged CCP1 or a catalytically dead version of the enzyme (H880S/E883Q, hereafter HS/EQ, [6]) in HeLa, HEK293T or CHL-1 cells; HeLa cells do not harbor significant levels of ∆Y-a-tubulin, whereas ∆Y-a-tubulin is readily detectable in HEK293T and CHL-1 cells (Figure 1A).”

We believe that these sentences provide the explanation requested by the reviewer, but thank him/her for requesting this clarification.
